# Fully Sparse 3D Object Detection

Lue Fan[1,2,3,4]    Feng Wang[5]    Naiyan Wang[5]    Zhaoxiang Zhang[1,2,3,6,✉]

[1]Institute of Automation, Chinese Academy of Sciences
[2]University of Chinese Academy of Sciences
[3]National Laboratory of Pattern Recognition, CASIA
[4]School of Future Technology, UCAS
[5]TuSimple    [6]Center for Artificial Intelligence and Robotics, HKISI_CAS
{fanlue2019, zhaoxiang.zhang}@ia.ac.cn    {feng.wff, winsty}@gmail.com

## Abstract

As the perception range of LiDAR increases, LiDAR-based 3D object detection becomes a dominant task in the long-range perception task of autonomous driving. The mainstream 3D object detectors usually build dense feature maps in the network backbone and prediction head. However, the computational and spatial costs on the dense feature map are quadratic to the perception range, which makes them hardly scale up to the long-range setting. To enable efficient long-range LiDAR-based object detection, we build a fully sparse 3D object detector (FSD). The computational and spatial cost of FSD is roughly linear to the number of points and independent of the perception range. FSD is built upon the general sparse voxel encoder and a novel sparse instance recognition (SIR) module. SIR resolves the issue of center feature missing, which hinders the design of the fully sparse architecture. Moreover, SIR avoids the time-consuming neighbor queries in previous point-based methods. We conduct extensive experiments on the large-scale Waymo Open Dataset to reveal the inner workings, and state-of-the-art performance is reported. To demonstrate the superiority of FSD in long-range detection, we also conduct experiments on Argoverse 2 Dataset, which has a much larger perception range ($200m$) than Waymo Open Dataset ($75m$). On such a large perception range, FSD achieves state-of-the-art performance and is $2.4\times$ faster than the dense counterpart. Our code is released at `https://github.com/TuSimple/SST`.

## 1  Introduction

Autonomous driving systems are eager for efficient long-range perception for downstream tasks, especially in high-speed scenarios. Current 3D LiDAR-based object detectors usually convert sparse features into dense feature maps for further feature extraction and prediction. For simplicity, we name the detectors utilizing dense feature maps as **dense detectors**. Dense detectors perform well on current popular benchmarks [31, 7, 2], where the perception range is relatively short (less than 75 meters). However, it is impractical to scale the dense detectors to the long-range setting (more than 200 meters, Fig. 2). In such settings, the computational and spatial complexity on dense feature maps is quadratic to the perception range. Fortunately, the sparsity of LiDAR point clouds also increases as the perception range extends (see Fig. 2), and the calculation on the

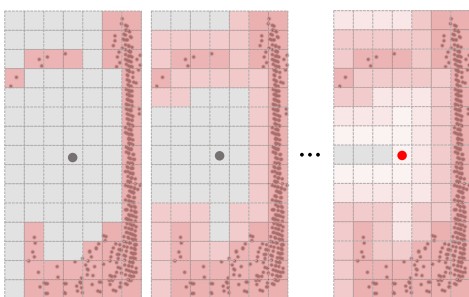

Figure 1: Illustration of **center feature missing** and **feature diffusion** on dense feature maps from Bird's Eye View. The empty instance center (red dot) is filled by the features diffused from occupied voxels (with LiDAR points), after several convolutions.

unoccupied area is essentially unnecessary. Given the inherent sparsity, an essential solution for efficient long-range detection is to remove the dense feature maps and make the network architectures *fully sparse*.

However, removing the dense feature map is non-trivial since it plays a critical role in current designs. Commonly adopted sparse voxel encoders [38, 5, 29] only extract the features on the non-empty voxels for efficiency. So without dense feature maps, the object centers are usually empty, especially for large objects. We name this issue as "**Center Feature Missing (CFM)**" (Fig. 1). Almost all popular voxel or pillar based detectors [28, 5, 42, 30, 38] in this field adopt center-based or anchor-based assignment since the center feature is the best representation of the whole object. However, CFM significantly weakens the representation power of the center voxels, even makes the center feature empty in some extreme cases like super large vehicles. Given this difficulty, many previous detectors [38, 42, 28, 5] have to convert sparse voxels to dense feature maps in Bird's Eye View after the sparse voxel encoder. Then they resolve the CFM issue by applying convolutions on the dense feature maps to diffuse features to instance centers, which we name as **feature diffusion** (Fig. 1).

To properly remove the dense feature map, we then investigate the purely point-based detectors because they are naturally fully sparse. However, two drawbacks limit the usage of point-based methods in the autonomous driving scenario. (1) Inefficiency: The time-consuming neighborhood query [24] is the long-standing difficulty to apply it to large-scale point cloud (more than 100K points). (2) Coarse representation: To reduce the computational overhead, point-based methods aggressively downsample the whole scene to a fixed number of points. The aggressive downsampling leads to inevitable information loss and insufficient recall of foreground objects [40, 43]. As a result, very few purely point-based detectors have reached state-of-the-art performance in the recent benchmarks with large-scale point clouds.

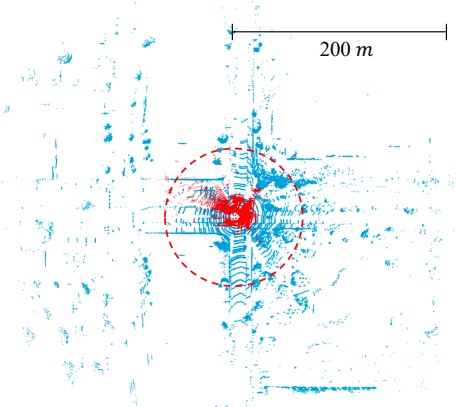

Figure 2: Short-range point clouds (red, from KITTI [7]) v.s. long-range point clouds (blue, from Argoverse 2 [37]). The radius of the red circle is 75 meters. The sparsity quickly increases as the range extends.

In this paper, we propose **Fully Sparse Detector (FSD)**, which takes the advantages of both sparse voxel encoder and point-based instance predictor. Since the central region might be empty, the detector has to predict boxes from other non-empty parts of instances. However, predicting the whole box from individual parts causes a large variance on the regression targets, making the results noisy and inconsistent. This motivates us to first group the points into an instance, then we further extract the instance-level feature and predict a single bounding box from the instance feature. To implement this principle, FSD first utilizes the sparse voxel encoder [38, 29, 5] to extract voxel features, then votes object centers based on these features as in VoteNet [23]. Then the *Instance Point Grouping (IPG)* module groups the voted centers into instances via Connected Components Labeling. After grouping, a point-based *Sparse Instance Recognition (SIR)* module extracts instance features and predicts the whole bounding boxes. As a point-based module, SIR has several desired properties. (1) Unlike previous point-based modules, SIR treats instances as groups, and does not apply the time-consuming neighborhood query for further grouping. (2) Similar to dynamic voxelization [46], SIR leverages *dynamic broadcast/pooling* for tensor manipulation to avoid point sampling or padding. (3) Since SIR covers the whole instance, it builds a sufficient receptive field regardless of the physical size of the instance. We list our contributions as follows.

- We propose the concept of Fully Sparse Detector (FSD), which is the essential solution for efficient long-range LiDAR detection. We further propose Sparse Instance Recognition (SIR) to overcome the issue of Center Feature Missing in sparse feature maps. Combining SIR with general sparse voxel encoders, we build an efficient and effective FSD implementation.

- FSD achieves state-of-the-art performance on the commonly used Waymo Open Dataset. Besides, we further apply our method to the recently released Argoverse 2 dataset to demonstrate the superiority of FSD in long-range detection. Given its challenging 200 meters perception range, FSD showcases state-of-the-art performance while being 2.4× faster than state-of-the-art dense detectors.

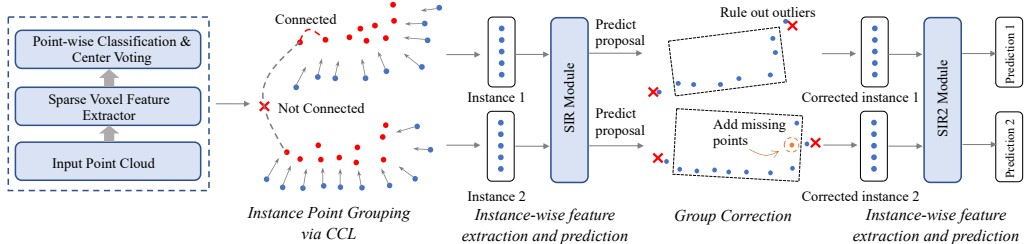

Figure 3: Overall architecture of FSD. For simplicity, we only use two instances to illustrate the pipeline. Red dots are the voted centers from each LiDAR point (blue dots). The SIR module and the SIR2 module all contain 3 SIR layers.

## 2 Related Work

**Voxel-based dense detectors** Pioneering work VoxelNet [45] uses dense convolution for voxel feature extraction. Although it achieves competitive performance, it is inefficient to apply dense convolution to 3D voxel representation. PIXOR [39] and PointPillars [13] adopt 2D dense convolution in Bird's Eye View (BEV) feature map achieving significant efficiency improvement. We name such detectors as **dense detectors** since they convert the sparse point cloud into dense feature maps.

**Voxel-based semi-dense detectors** Different from the dense detectors, **semi-dense detectors** incorporate both sparse features and dense features. SECOND [38] adopts sparse convolution to extract the sparse voxel features in 3D space, which then are converted to dense feature maps in BEV to enlarge the receptive field and integrate with 2D detection head [19, 25, 44]. Based on SECOND-style semi-dense detectors, many methods attach a second stage for fine-grained feature extraction and proposal refinement [29, 28, 30, 3]. It is noteworthy that the semi-dense detector is hard to be trivially lifted to the fully sparse detector since it will face the issue of Center Feature Missing, as we discussed in Sec. 1.

**Point-based sparse detectors** The purely point-based detectors are born to be fully sparse. PointR-CNN [27] is the pioneering work to build the purely point-based detector. 3DSSD [40] accelerates the point-based method by removing the feature propagation layer and refinement module. VoteNet [23] first makes a center voting and then generates proposals from the voted center achieving better precision. Albeit many methods have tried to accelerate the point-based method, the time-consuming neighborhood query is still unaffordable in large-scale point clouds (more than 100k points per scene). So current benchmarks [31, 2] with large-scale point clouds are dominated by voxel-based dense/semi-dense detectors [11, 30, 15].

## 3 Methodology

### 3.1 Overall Architecture

Following the motivation of instances as groups, we have four steps to build the fully sparse detector (FSD): 1) We first utilize a sparse voxel encoder [5, 29, 38] to extract voxel features and vote object centers(Sec. 3.2). 2) Instance Point Grouping groups foreground points into instances based on the voting results (Sec. 3.2). 3) Given the grouping results, Sparse Instance Recognition (SIR) module extracts instance/point features and generates proposals (Sec. 3.3). 4) The proposals are utilized to correct the point grouping and refine the proposals via another SIR module (Sec. 3.4).

### 3.2 Instance Point Grouping

**Classification and Voting** We first extract voxel features from the point cloud with a sparse voxel encoder. Although FSD is not restricted to a certain sparse voxel encoder, we utilize sparse attention block in SST [5] due to its demonstrated effectiveness. Then we build point features by concatenating voxel features and the offsets from points to their corresponding voxel centers. These point features are passed into two heads for foreground classification and center voting. The voting is similar to VoteNet [23], where the model predicts the offsets from foreground points to corresponding object

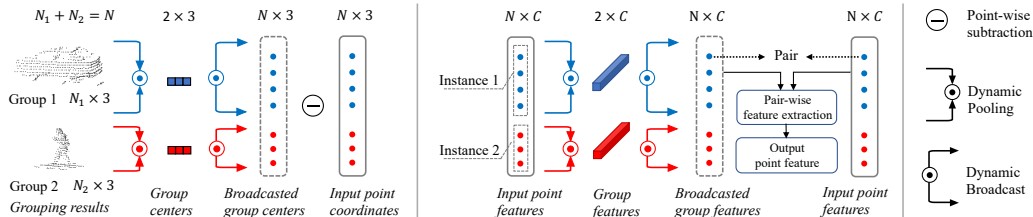

Figure 4: Illustration of building instance-level point operators with dynamic broadcast/pooling. Best viewed in color. Left: calculating center-to-neighbor offsets given raw point clouds. Right: updating point features. Note that the operation is parallel among all instances.

centers. L1 loss [25] and Focal Loss [18] are adopted as voting loss $L_{vote}$ and semantic classification loss $L_{sem}$.

**Connected Components Labeling (CCL)** To group points into instances, we regard all the predicted centers (red dots in Fig. 3) as vertices in a graph. Two vertices are connected if their distance is smaller than a certain threshold. Then a connected component in this graph can be viewed as an instance, and all points voted to this connected component share a group ID. Unlike the ball query in VoteNet, our CCL-based grouping greatly avoids fragmented instances. Although there are many elaborately designed instance grouping methods [12, 34, 10], we opt for the simple CCL because it is adequate in our design and can be implemented by the efficient depth-first search.

### 3.3 Sparse Instance Recognition

#### 3.3.1 Preliminaries: Dynamic Broadcast/Pooling

Given N points belong to M groups, we define their corresponding group ID array as $I$ in shape of $[\text{N},\,]$ and their feature array as $F$ in shape of $[\text{N},\text{C}]$, where C is the feature dimensions. $F^{(i)}$ is the feature array of points belonging to the $i$-th group. Dynamic pooling aggregates each $F^{(i)}$ into one group feature $\boldsymbol{g}_i$ of shape $[\text{C},\,]$. Thus we have $\boldsymbol{g}_i = p(F^{(i)})$, where $p$ is a symmetrical pooling function. The dynamic pooling on all group features $G$ of shape $[\text{M},\text{C}]$ is formulated as $G = p(F, I)$. The dynamic broadcast can be viewed as the inverse operation to dynamic pooling, which broadcasts $\boldsymbol{g}_i$ to all the points in the $i$-th group. Since the broadcasting is essentially an indexing operation, we use the indexing notation $[\,]$ to denote it as $G[I]$, which is in shape of $[\text{N},\text{C}]$. Dynamic broadcast/pooling is very efficient because it can be implemented with high parallelism on modern devices and well fits the sparse data with dynamic size.

The prerequisite of dynamic broadcast/pooling is that each point *uniquely* belongs to a group. In other words, groups should not overlap with each other. Thanks to the motivation of instances as groups, the groups in 3D space do not overlap with each other naturally.

#### 3.3.2 Formulation of Sparse Instance Recognition

After grouping points into instances in Sec. 3.2, we can directly extract instance features by some basic point-based networks like PointNet, DGCNN, etc. There are three elements to define a basic point-based module: *group center*, *pair-wise feature* and *group feature aggregation*.

**Group center** The group center is the representative point of a group. For example, in the ball query, it is the local origin of the sphere. In SIR, the group center is defined as the centroid of all voted centers in a group.

**Pair-wise feature** defines the input for per point feature extraction. SIR adopts two kinds of features: 1) relative coordinate between group center and each point, 2) feature concatenation between group and each neighbor point. Taking feature concatenation as example and using the notations in 3.3.1, the pair-wise feature can be denoted as $\text{CAT}(F, G[I])$, where CAT is channel concatenation.

**Group feature aggregation** In a group, a pooling function is used to aggregate neighbor features. SIR applies dynamic pooling to aggregate feature array $F$. Following the notations in 3.3.1, we have $G = p(F, I)$, where $G$ is the aggregated group features.

**Integration** Combining the three basic elements, we could build many variants of point-based operators, such as PointNet [22], DGCNN [35], Meta-Kernel [4], etc. Fig. 4 illustrates the basic idea of how to build an instance-level point operator with dynamic broadcast/pooling. In our design, we adopt the formulation of VFE [45] as the basic structure of SIR layers, which is basically a two-layer PointNet. In the $l$-th layer of SIR module, given the input point-wise feature array $F_l$, point coordinates array $X$, the voted center $X'$ and group ID array $I$, the output of $l$-th layer can be formulated as:

$$F_l' = \texttt{LinNormAct}\left(\texttt{CAT}\left(F_l, X - p_{\text{avg}}(X', I)[I]\right)\right),\tag{1}$$

$$F_{l+1} = \texttt{LinNormAct}\left(\texttt{CAT}\left(F_l', p_{\text{max}}(F_l', I)[I]\right)\right),\tag{2}$$

where $\texttt{LinNormAct}$ is a fully-connected layer followed by a normalization layer [33] and an activation function [9]. The $p_{\text{avg}}$ and the $p_{\text{max}}$ are average-pooling and max-pooling function, respectively. The output $F_{l+1}$ can be further used as the input of the next SIR layer, so our SIR module is a stack of a couple of basic SIR layers.

### 3.3.3 Sparse Prediction

With the formulation in Eqn. 1 and Eqn. 2, SIR extracts features of all instances dynamically in parallel. And then SIR makes **sparse prediction** for all groups. In contrast to two-stage sparse prediction, our proposals (i.e., groups) do not overlap with each other. Unlike one-stage dense prediction, we only generate a single prediction for a group. Sparse prediction avoids the difficulty of label assignment in dense prediction when the center feature is missing, because there is no need to attach anchors or anchor points to non-empty voxels. It is noteworthy that the fully sparse architecture may face a severe imbalance problem: short-range objects contain much more points than long-range objects. Some methods [1, 4] use hand-crafted normalization factors to mitigate the imbalance. Instead, SIR avoids the imbalance because it only generates a single prediction for a group regardless of the number of points in the group.

Specifically, for each SIR layer, there is a $G_l = p_{max}(F_l', I)$ in Eqn. 2, which can be viewed as the group features. We concatenate all $G_l$ from each SIR layer in channel dimension and use the concatenated group features to predict bounding boxes and class labels via MLPs. All the groups whose centers fall into ground-truth boxes are positive samples. For positive samples, the regression branch predicts the offsets from group centers to ground-truth centers and object sizes and orientations. L1 loss [25] and Focal Loss [18] are adopted as regression loss $L_{reg}$ and classification loss $L_{cls}$, respectively.

## 3.4 Group Correction

There is inevitable incorrect grouping in the Instance Point Grouping module. For example, some foreground points may be missed, or some groups may be contaminated by background clutter. So we leverage the bounding box proposals from SIR to correct the grouping. The points inside a proposal belong to a corrected group regardless of their previous group IDs. After correction, we apply an additional SIR to these new groups. To distinguish it from the first SIR module, we denote the additional SIR module as SIR2.

SIR2 predicts box residual from the proposal to its corresponding ground-truth box, following many two-stage detectors. To make SIR2 aware of the size and location of a proposal, we adopt the offsets from inside points to proposal boundaries as extra point features following [16]. The regression loss is denoted as $L_{res} = L1(\Delta_{res}, \widehat{\Delta_{res}})$, where $\Delta_{res}$ is the ground-truth residual and $\widehat{\Delta_{res}}$ is the predicted residual. Following previous methods [29, 28], the 3D Intersection over Union (IoU) between the proposal and ground-truth serves as the soft classification label in SIR2. Specifically, the soft label $q$ is defined as $q = \min(1, \max(0, 2IoU - 0.5))$, where $IoU$ is the IoU between proposals and corresponding ground-truth. Then cross entropy loss is adopted to train the classification branch, denoted as $L_{iou}$. Taking all the loss functions in grouping (Sec. 3.2) and sparse prediction into account, we have

$$L_{total} = L_{sem} + L_{vote} + L_{reg} + L_{cls} + L_{res} + L_{iou},\tag{3}$$

where we omit the normalization factors for simplicity.

### 3.5 Discussion

The center voting in FSD is inspired by VoteNet [23], while FSD has two essential differences from VoteNet.

- After voting, VoteNet simply aggregates features around the voted centers without further feature extraction. Instead, FSD builds a highly efficient SIR module taking advantage of dynamic broadcast/pooling for further instance-level feature extraction. Thus, FSD extracts more powerful instance features, which is experimentally demonstrated in Sec. 4.6.

- VoteNet is a typical point-based method. As we discussed in Sec. 1, it aggressively down-samples the whole scene to a fixed number of points for efficiency, causing inevitable information loss. Instead, the dynamic characteristic and efficiency of SIR enable fine-grained point feature extraction from any number of input points without any downsampling. In Sec. 4.6, we showcase the efficiency of our design in processing large-scale point clouds and the benefits from fine-grained point representation.

## 4 Experiments

### 4.1 Setup

**Dataset: Waymo Open Dataset (WOD)**    We conduct our main experiments on WOD [31]. WOD is currently the largest and most trustworthy benchmark for LiDAR-based 3D object detection. WOD contains 1150 sequences (more than 200K frames), 798 for training, 202 for validation, and 150 for test. The detection range in WOD is 75 meters (cover area of $150m \times 150m$).

**Dataset: Argoverse 2 (AV2)**    We further conduct long-range experiments on the recently released Argoverse 2 dataset [37] to demonstrate the superiority of FSD in long-range detection. AV2 has a similar scale to WOD, and it contains 1000 sequences in total, 700 for training, 150 for validation, and 150 for test. In addition to *average precision* (AP), AV2 adopts a *composite score* as evaluation metric, which takes both AP and localization errors into account. The perception range in AV2 is 200 meters (cover area of $400m \times 400m$), which is much larger than WOD. Such a large perception range leads to a huge memory footprint for dense detectors.

**Model Variants**    To demonstrate the generality of SIR, we build two FSD variants. $FSD_{sst}$ adopts the emerging single stride sparse transformer [5] as sparse voxel feature extractor. $FSD_{spconv}$ is built upon sparse convolution based U-Net in PartA2 [29]. Unless otherwise specified, we use $FSD_{sst}$ in the experiments.

**Implementation Details**    We use 4 sparse regional attention blocks [5] in SST as our voxel feature extractor. The SIR module and SIR2 module consist of 3 and 6 SIR layers, respectively. A SIR layer is defined by Eqn. 1 and Eqn. 2. Our SST-based model converges much faster than SST, so we train our models for 6 epochs instead of the $2\times$ schedule (24 epochs) in SST. For $FSD_{spconv}$, in addition to the 6-epoch schedule, we adopt a longer schedule (12 epochs) for better performance. Different from the default setting in MMDetection3D, we decrease the number of pasted instances in the CopyPaste augmentation, to prevent FSD from overfitting.

### 4.2 Comparison to State-of-the-art Methods

We first compare FSD with state-of-the-art detectors and our baseline in Table 1. FSD achieves the state-of-the-art performance among all the mainstream detectors. Thanks to the fine-grained feature extraction in SIR, FSD also obtains exciting performance on *Pedestrian* class and *Cyclist* class with single-frame point clouds.

### 4.3 Study of Treatments to Center Feature Missing

In what follows, we conduct experiments on WOD to elaborate the issue of **Center Feature Missing (CFM)**. We first build several models with different characteristics. Note that all the following models adopt the same voxelization resolution, so they face the same degree of CFM at the beginning.

- **$FSD_{plain}$**: After the sparse voxel encoder, $FSD_{plain}$ directly predicts the box from each voxel. The voxels inside ground-truth boxes are assigned *positive*. Although $FSD_{plain}$ uses the most

Table 1: Performances on the Waymo Open Dataset validation split. All models only take single-frame point cloud as input without any test-time augmentations or model ensemble. All classes are trained in a single model in FSD. Different from the default CopyPaste in MMDetection3D, we decrease the number of pasted instances to prevent overfitting. †: Longer schedule (12 epochs).

| Methods | mAP/mAPH L2 | *Vehicle* 3D AP/APH L1 | L2 | *Pedestrian* 3D AP/APH L1 | L2 | *Cyclist* 3D AP/APH L1 | L2 |
|---|---|---|---|---|---|---|---|
| SECOND [38] | 61.0/57.2 | 72.3/71.7 | 63.9/63.3 | 68.7/58.2 | 60.7/51.3 | 60.6/59.3 | 58.3/57.0 |
| MVF [46] | -/- | 62.9/- | -/- | 65.3/- | -/- | -/- | -/- |
| AFDet [6] | -/- | 63.7/- | -/- | -/- | -/- | -/- | -/- |
| Pillar-OD [36] | -/- | 69.8/- | -/- | 72.5/- | -/- | -/- | -/- |
| RangeDet [4] | 65.0/63.2 | 72.9/72.3 | 64.0/63.6 | 75.9/71.9 | 67.6/63.9 | 65.7/64.4 | 63.3/62.1 |
| PointPillars [13] | 62.8/57.8 | 72.1/71.5 | 63.6/63.1 | 70.6/56.7 | 62.8/50.3 | 64.4/62.3 | 61.9/59.9 |
| Voxel RCNN [3] | -/- | 75.6/- | 66.6/- | -/- | -/- | -/- | -/- |
| RCD [1] | -/- | 69.0/68.5 | -/- | -/- | -/- | -/- | -/- |
| VoTr-TSD [21] | -/- | 74.9/74.3 | 65.9/65.3 | -/- | -/- | -/- | -/- |
| LiDAR-RCNN [16] | 65.8/61.3 | 76.0/75.5 | 68.3/67.9 | 71.2/58.7 | 63.1/51.7 | 68.6/66.9 | 66.1/64.4 |
| Pyramid RCNN [20] | -/- | 76.3/75.7 | 67.2/66.7 | -/- | -/- | -/- | -/- |
| Voxel-to-Point [14] | -/- | 77.2/- | 69.8/- | -/- | -/- | -/- | -/- |
| 3D-MAN [41] | -/- | 74.5/74.0 | 67.6/67.1 | 71.7/67.7 | 62.6/59.0 | -/- | -/- |
| M3DETR [8] | 61.8/58.7 | 75.7/75.1 | 66.6/66.0 | 65.0/56.4 | 56.0/48.4 | 65.4/64.2 | 62.7/61.5 |
| Part-A2-Net [29] | 66.9/63.8 | 77.1/76.5 | 68.5/68.0 | 75.2/66.9 | 66.2/58.6 | 68.6/67.4 | 66.1/64.9 |
| CenterPoint-Pillar [42] | -/- | 76.1/75.5 | 68.0/67.5 | 76.1/65.1 | 68.1/57.9 | -/- | -/- |
| CenterPoint-Voxel [42] | 69.8/67.6 | 76.6/76.0 | 68.9/68.4 | 79.0/73.4 | 71.0/65.8 | 72.1/71.0 | 69.5/68.5 |
| IA-SSD [43] | 62.3/58.1 | 70.5/69.7 | 61.6/61.0 | 69.4/58.5 | 60.3/50.7 | 67.7/65.3 | 65.0/62.7 |
| PV-RCNN [28] | 66.8/63.3 | 77.5/76.9 | 69.0/68.4 | 75.0/65.6 | 66.0/57.6 | 67.8/66.4 | 65.4/64.0 |
| RSN [32] | -/- | 75.1/74.6 | 66.0/65.5 | 77.8/72.7 | 68.3/63.7 | -/- | -/- |
| SST_TS [5] | -/- | 76.2/75.8 | 68.0/67.6 | 81.4/74.0 | 72.8/65.9 | -/- | -/- |
| SST [5] | 67.8/64.6 | 74.2/73.8 | 65.5/65.1 | 78.7/69.6 | 70.0/61.7 | 70.7/69.6 | 68.0/66.9 |
| AFDetV2 [11] | 71.0/68.8 | 77.6/77.1 | 69.7/69.2 | 80.2/74.6 | 72.2/67.0 | 73.7/72.7 | 71.0/70.1 |
| PillarNet-34 [26] | 71.0/68.5 | 79.1/78.6 | **70.9/70.5** | 80.6/74.0 | 72.3/66.2 | 72.3/71.2 | 69.7/68.7 |
| PV-RCNN++ [30] | 68.4/64.9 | 78.8/78.2 | 70.3/69.7 | 76.7/67.2 | 68.5/59.7 | 69.0/67.6 | 66.5/65.2 |
| PV-RCNN++(center) [30] | 71.7/69.5 | **79.3**/78.8 | 70.6/70.2 | 81.3/76.3 | 73.2/68.0 | 73.7/72.7 | 71.2/70.2 |
| FSD$_{spconv}$ (ours) | 71.9/69.7 | 77.8/77.3 | 68.9/68.5 | 81.9/76.4 | 73.2/68.0 | 76.5/75.2 | 73.8/72.5 |
| FSD$_{sst}$ (ours) | 71.5/69.2 | 76.8/76.3 | 67.9/67.5 | 81.3/75.3 | 72.5/67.0 | **77.2/76.0** | **74.4**/73.2 |
| FSD$_{spconv}$ (ours) † | **72.9/70.8** | 79.2/**78.8** | 70.5/70.1 | **82.6/77.3** | **73.9/69.1** | 77.1/**76.0** | **74.4/73.3** |

straightforward solution for CFM, it suffers from the large variance of regression targets and low-quality predictions from hard voxels.

- **SST**$_{center}$: It replaces the anchor-based head in SST with CenterHead [44, 42]. Based on sparse voxel encoder, SST$_{center}$ converts sparse voxels into dense feature maps and applies several convolutions to diffuse features to the empty object centers as in Fig. 1. Then it makes predictions from the diffused center feature.

- **FSD**$_{nogc}$: It removes the group correction and SIR2 module in FSD.

- **CenterPoint-PP**: It does not resort to any sparse voxel encoders. Instead, it applies multiple dense convolutions soon after voxelization for feature diffusion, greatly eliminating CFM. It also is equipped with CenterHead avoiding large variance of regression targets.

**Experiments and analyses** There is usually a quite large unoccupied area around the centers of large vehicles. Thus the performance of large vehicles is an appropriate indicator that reveals the effect of CFM. So we build a customized evaluation tool, which breaks down the object length following the COCO evaluation [17]. Then we use it to evaluate the performance of vehicles with different lengths. Table 2 shows the results, and we list our findings as follows.

Table 2: Vehicle detection with vehicle length breakdown. †: re-implemented ourselves. *: official Waymo L2 overall metric. Arrows indicate the performance changes from SST$_{center}$.

| Methods | Vehicle length (m) [0, 4) | [4, 8) | [8, 12) | [12, +∞) | Official* |
|---|---|---|---|---|---|
| CenterPoint-PP† | 34.3 | 69.3 | 42.0 | 43.6 | 66.2 |
| FSD$_{plain}$ | 32.2 | 64.6 | 41.3 | 42.2 | 62.3 |
| SST$_{center}$ [5] | 36.0 | 69.4 | 33.7 | 30.5 | 66.3 |
| FSD$_{nogc}$ | 33.5 ↓ 2.5 | 68.2 ↓ 1.2 | 47.7 ↑ 14.0 | 47.9 ↑ 17.4 | 65.2 ↓ 1.1 |
| FSD | 36.7 ↑ 0.7 | 71.0 ↑ 1.6 | 51.3 ↑ 17.6 | 53.7 ↑ 23.2 | 69.3 ↑ 3.0 |

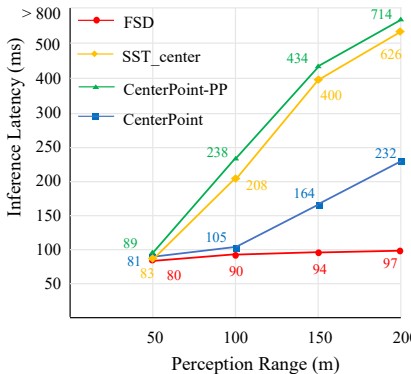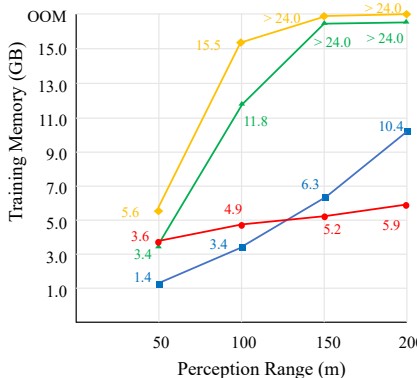

Figure 5: **Memory footprints and inference latency in different perception ranges.** We use FSD$_{sst}$ (Sec. 4.1) here. Statistics are obtained on a single 3090 GPU with batch size 1. Inference latency is evaluated by the standard benchmark script in MMDetection3D without any test-time optimization. CenterPoint-PP and SST$_{center}$ are defined in Sec. 4.3. Best viewed in color.

- Comparing FSD$_{plain}$ with SST$_{center}$, they share the same attention-based sparse voxel encoder. However, the trend is totally opposite w.r.t vehicle size. With feature diffusion, SST$_{center}$ attains much worse performance than FSD$_{plain}$ on large vehicles. It suggests feature diffusion is a sub-optimal solution for CFM in the case of large objects. For those large objects, the features may not be diffused to the centers or the diffused features are too weak to make accurate predictions.

- However, FSD$_{plain}$ obtains the worst performance among all detectors on vehicles with normal sizes. Note that the CFM issue is minor for the normal size vehicles. So, in this case, the center-based assignment in SST$_{center}$ shows its superiority to the assignment in FSD$_{plain}$. It suggests the solution for CFM in FSD$_{plain}$ is also sub-optimal, even if it achieves better performance in large objects.

- Comparing FSD$_{nogc}$ with SST$_{center}$, they share the same sparse voxel encoder while FSD$_{nogc}$ replaces the dense part in SST$_{center}$ with SIR. The huge improvements of FSD$_{nogc}$ on large vehicles fairly reveal that SIR effectively resolves CFM and is better than feature diffusion.

- CenterPoint-PP suffers much less from CFM because it leverages dense feature maps from very beginning of the network. It is also equipped with the advanced center-based assignment. Even so, FSD$_{nogc}$ and FSD still outperform CenterPoint-PP, especially on large vehicles.

Table 3: Performance in Argoverse 2 validation split. †: provided by authors of AV2 dataset. ‡: Weak CopyPaste augmentation for preventing overfitting (one instance per class). ∗: re-implemented by ourselves. C-Barrel: construction barrel. MPC-Sign: mobile pedestrian crossing sign. A-Bus: articulated bus. C-Cone: construction cone. V-Trailer: vehicular trailer. We omit the results of dog, wheelchair and message board trailer because these categories contain very few instances. The average results take all categories into account, including the omitted categories. We mark the categories attaining notable improvements in ***bold***.

| Methods | Average | Vehicle | Bus | Pedestrian | Stop Sign | Box Truck | Bollard | C-Barrel | Motorcyclist | MPC-Sign | Motorcycle | Bicycle | A-Bus | School Bus | Truck Cab | C-Cone | V-Trailer | Sign | Large Vehicle | Stroller | Bicyclist |
|---|---|---|---|---|---|---|---|---|---|---|---|---|---|---|---|---|---|---|---|---|---|
| *Precision* | | | | | | | | | | | | | | | | | | | | | |
| CenterPoint† [42] | 13.5 | 61.0 | 36.0 | 33.0 | 28.0 | 26.0 | 25.0 | 22.5 | 16.0 | 16.0 | 12.5 | 9.5 | 8.5 | 7.5 | 8.0 | 8.0 | 7.0 | 6.5 | 3.0 | 2.0 | 14 |
| CenterPoint* | 22.0 | 67.6 | 38.9 | 46.5 | 16.9 | 37.4 | 40.1 | 32.2 | 28.6 | 27.4 | 33.4 | 24.5 | 8.7 | 25.8 | 22.6 | 29.5 | 22.4 | 6.3 | 3.9 | 0.5 | 20.1 |
| FSD | 24.0 | 67.1 | 39.8 | 57.4 | 21.3 | 38.3 | 38.3 | 38.1 | 30.0 | 23.6 | 38.1 | 25.5 | 15.6 | 30.0 | 20.1 | 38.9 | 23.9 | 7.9 | 5.1 | 5.7 | 27.0 |
| FSD$_{spconv}$‡ | 28.2 | 68.1 | 40.9 | 59.0 | 29.0 | 38.5 | 41.8 | 42.6 | 39.7 | 26.2 | 49.0 | 38.6 | 20.4 | 30.5 | 14.8 | 41.2 | 26.9 | 11.9 | 5.9 | 13.8 | 33.4 |
| *Composite Score* | | | | | | | | | | | | | | | | | | | | | |
| CenterPoint* | 17.6 | 57.2 | 32.0 | 35.7 | 13.2 | 31.0 | 28.9 | 25.6 | 22.2 | 19.1 | 28.2 | 19.6 | 6.8 | 22.5 | 17.4 | 22.4 | 17.2 | 4.8 | 3.0 | 0.4 | 16.7 |
| FSD | 19.1 | 56.0 | 33.0 | 45.7 | 16.7 | 31.6 | 27.7 | 30.4 | 23.8 | 16.4 | 31.9 | 20.5 | 12.0 | 25.6 | 15.9 | 29.2 | 18.1 | 6.4 | 3.8 | 4.5 | 22.1 |
| FSD$_{spconv}$‡ | 22.7 | 57.7 | 34.2 | 47.5 | 23.4 | 31.7 | 30.9 | 34.4 | 32.3 | 18.0 | 41.4 | 32.0 | 15.9 | 26.1 | 11.0 | 30.7 | 20.5 | 9.5 | 4.4 | 11.5 | 28.0 |

## 4.4 Long-range Detection

Several widely adopted 3D detection benchmarks [31, 7, 2] have relatively short perception range. To unleash the potential of FSD, we conduct long-range detection experiments on the recently released

Argoverse 2 dataset (AV2), with a perception range of 200 meters. In addition, AV2 contains objects in 30 classes, facing the challenging long-tail issue.

**Main results**    We first list the main results of FSD on AV2 in Table 3. The authors of AV2 provide a baseline CenterPoint model, but the results are mediocre. To make a fair comparison, we re-implement a stronger CenterPoint model on the AV2 dataset. The re-implemented CenterPoint adopts the same training scheme with FSD, including ground-truth sampling to alleviate the long-tail issue. FSD outperforms CenterPoint in the average metric. It is noteworthy that FSD significantly outperforms CenterPoint in some tiny objects (e.g., Pedestrian, Construction Cone) as well as some objects with extremely large sizes (e.g., Articulated Bus, School Bus). We owe this to the virtue of instance-level fine-grained feature extraction in SIR.

**Range Scaling**    To demonstrate the efficiency of FSD in long-range detection, we depict the trend of training memory and inference latency of three detectors when the perception range increases in Fig. 5. Fig. 5 shows dramatic latency/memory increase when applying dense detectors to larger perception ranges. Designed to be fully sparse, the resource needed for FSD is roughly linear to the number of input points, so its memory and latency only slightly increase as the perception range extends.

## 4.5    More Sparse Scenes

Argoverse 2 dataset provides a highly reliable HD map, which could be utilized as a prior to remove uninterested regions making the scene more sparse. Thus we proceed with experiments removing some uninterested regions to show the advantages of FSD in more sparse scenarios. The results are summarized in Table 4. FSD has

Table 4: Performance with different detection areas. †: Region of Interest is defined by the HD map in AV2 dataset.

| | FSD | | | CenterPoint | | |
|---|---|---|---|---|---|---|
| | Mem. | Latency(ms) | mAP | Mem. | Latency(ms) | mAP |
| all | 5.9 | 97 | 24.0 | 10.4 | 232 | 22.0 |
| only RoI† | 3.2 ↓ 45.8% | 81↓ 16.5% | 23.2 | 9.9↓ 4.8% | 227↓ 2.2% | 21.5 |
| w/o ground | 2.3 ↓ 61.0% | 74↓ 25.8% | 21.0 | 9.7↓ 6.7% | 217↓ 6.4% | 19.8 |

a significantly lower memory footprint and latency with an acceptable precision loss after removing the uninterested regions. On the contrary, the efficiency improvement of CenterPoint is minor. It reveals that FSD benefits more from the increase of data sparsity, which is another advantage of the fully sparse architecture.

## 4.6    Ablation Study

**Effectiveness of Components**    In addition to $FSD_{plain}$ and $FSD_{nogc}$ (Sec. 4.3), we also degrade FSD to $FSD_{agg}$ to understand the mechanism of FSD. In $FSD_{agg}$, we aggregate grouped point features by dynamic pooling and then directly make predictions from the pooled features, after Instance Point Grouping. $FSD_{agg}$ is similar to the way in VoteNet [23] as we discussed

Table 5: Ablation of design factors in SIR. Performances are evaluated on Waymo validation split.

| | Grouping | SIR | Group Correction | L2 3D APH | | |
|---|---|---|---|---|---|---|
| | | | | Vehicle | Pedestrian | Cyclist |
| $FSD_{plain}$ | | | | 62.29 | 64.31 | 64.49 |
| $FSD_{agg}$ | ✓ | | | 63.13 | 65.13 | 64.52 |
| $FSD_{nogc}$ | ✓ | ✓ | | 65.20 | 67.39 | 67.78 |
| FSD | ✓ | ✓ | ✓ | 69.30 | 69.30 | 69.60 |

in Sec. 3.5. Thus, $FSD_{agg}$ can explicitly leverage instance-level features other than the point-level features in $FSD_{plain}$. However, $FSD_{agg}$ can not take advantage of further point feature extraction in SIR. As can be seen in Table 5, the improvement is limited if we only apply grouping without SIR. The combination of grouping and SIR attain notable improvements.

**Downsampling in SIR**    The efficiency of SIR makes it feasible to extract fine-grained point features without any point downsampling. This is another notable difference between FSD and VoteNet. To demonstrate the superiority, we apply voxelization on the raw points before SIR module and treat the centroids of voxels as downsampled points. We conduct experiments on AV2 dataset because it contains a couple of

Table 6: Performances with different representation granularity. †: Latency of SIR module.

| Voxel size | CC | Bollard | AP Bicyclist | Stop Sign | Latency (ms)† |
|---|---|---|---|---|---|
| 30cm | 35.4 | 36.5 | 24.6 | 18.3 | 3.5 |
| 20cm | 37.3 | 37.3 | 26.4 | 20.0 | 4.1 |
| 10cm | 38.9 | 38.3 | 27.0 | 21.3 | 4.5 |
| Point | 39.3 | 38.6 | 27.1 | 21.5 | 6.3 |

categories in a tiny size, which may be sensitive to downsampling. As expected, small objects have notable performance loss when adopting downsampling, and we list some of them in Table 6. We

also evaluate the inference latency of the SIR module on 3090 GPU. As can be seen, compared with the overall latency (97ms, Fig. 5), the SIR module is highly efficient.

## 5 Conclusion

This paper proposes FSD, a fully sparse 3D object detector, aiming for efficient long-range object detection. FSD utilizes a highly efficient point-based Sparse Instance Recognition module to solve the center feature missing in fully sparse architecture. FSD achieves not only competitive performance on the widely-used Waymo Open Dataset, but also state-of-the-art performance in the long-range Argoverse 2 dataset with a much faster inference speed than previous detectors.

**Limitation**  A more elaborately designed grouping strategy may help with performance improvements. However, it is beyond our design goal in this paper, and we will pursue it in future work.

## 6 Acknowledgements

This work was supported in part by the Major Project for New Generation of AI (No.2018AAA0100400), the National Natural Science Foundation of China (No. 61836014, No. U21B2042, No. 62072457, No. 62006231), and in part by the TuSimple Collaborative Research Project.

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
