# OpenReview forum: "Fully Sparse 3D Object Detection"
_NeurIPS.cc/2022/Conference — NeurIPS 2022 Accept_

### Official Review · Reviewer_UjRj · 2022-06-25

**Rating:** 7
**Confidence:** 3
**Soundness:** 2 fair
**Presentation:** 2 fair
**Contribution:** 2 fair

**Summary:**

The authors propose a novel 3D lidar point cloud detection method. the method is based on
a sparse instance recognition module which groups points into features, making the feature
extraction and bounding boxes regression for detection more efficient than other point-based
methods.

**Questions:**

However, it would be nice to have a quick explanation for the metric used as well as some qualitative
visual results, especially for the "failing"/less good results compared to other methods.
Also perhaps for the memory and latency performances, having more of the SOTA methods for comparison
would bring more value to the discussion.
Finally, even if the paper is more focused on long and sparse lidar data compared to other methods.
It would be interesting for a small comparison on the KITTI dataset, to see if the method
still performs correctly on shorter-range point clouds.

**Ethics Review Area:**

["I don’t know"]

**Limitations:**

The limitation mentioned by the authors is found in the grouping strategy employed.
I agree that a comparison with another grouping method would be out of the scope of the paper, but then perhaps an extra comparison with the grouping method used by another method would be interesting? (pointnet++ multi-scale grouping module for example).

**Strengths And Weaknesses:**

The motivation and goals of the work are well explained and clear.
The experimentation section is good and the ablation study brings interesting conclusions.

The paper has some little missing information listed in the questions below.

---

> ### Author Response · Authors · 2022-08-02
> **Response to R#3**
>
> We really appreciate your affirmation of our work and the constructive advice.
> ## Q1: Metrics and Qualitative Results.
> We are sorry for the lack of explanation for some metrics.
> We guess some readers may be unfamiliar with the metrics used in Argoverse 2 dataset.
> Here we provide the detailed official explanations ([link](https://github.com/argoai/av2-api/blob/main/src/av2/evaluation/detection/eval.py)).
> And we will make a quick explanation in the revised version as the reviewer kindly suggested.
> As for failure cases, a common one is that two close objects are grouped together by the CCL (Sec. 3.2) if the distance threshold in CCL is not suitable, leading to a false negative.
> We provide a rare case in the **Section C of our revised supplementary materials** for illustration. Fortunately, it could be solved with a different grouping method.
>
>
> ## Q2: More Comparisons on Memory and Latency
> It is a valuable suggestion, and we discuss this in two aspects.
>
> (1) The compared method CenterPoint is one of the most efficient SOTA methods.
> Besides, the official reported latency of top-performing PVRCNN++ is around 100ms (10 FPS), which is still slower than FSD (~ 60ms). Due to the time limit to training PVRCNN++ (no released model weights), we will make a stricter comparison in the future.
>
> (2) Almost all other SOTA methods (e.g., CenterPoint, PVRCNN++) adopt dense feature maps in their backbone, and the computational costs on dense feature maps are quadratic to the perception range.
> So it could be expected that FSD will be more efficient than these SOTA methods in the long-range setting.
>
> ## Q3: Short-range Performance
> We appreciate this very thoughtful consideration.
> We first humbly clarify that the range in the KITTI dataset is only a little shorter than Waymo Open Dataset ( $70m$ v.s. $75m$).
> To resolve the reviewer's concern, we re-benchmark the performance in different range intervals with the official Waymo evaluation tool.
> And the performances are shown in the following table.
>
> | Method | Veh. L1 AP (0m-30m) | Veh. L1 AP (30m-50m) | Ped. L1 AP (0m-30m) | Ped. L1 AP (30m-50m) | Cyc. L1 AP (0m-30m) | Cyc. L1 AP (30m-50m) |
> | --- | --- | --- | --- | --- |--- | --- |
> | PVRCNN++(center) | 93.3 | 78.1 | 84.9 | 79.7 | 83.7 | 68.9 |
> | FSD | 93.3 | 78.4 | 87.0 | 82.6 | 86.1 | 70.9 |
>
> Compared with the top-performing PVRCNN++(center), FSD shows comparable or better performance in the shorter range.
>
> ## Q4: Other Grouping Methods
> We agree with the reviewer that adopting other grouping methods would be interesting. However, the multi-scale grouping causes overlaps between different scale, which is incompatible with our current pipeline. Due to the time limit, we follow the suggestion of the reviewer, while simplifying the multi-scale grouping to single-scale grouping (SSG).
> The basic idea is to apply Farthest Point Sampling in voted centers and then use ball query in these sampled centers to discover the groups. The results are shown in the following table.
>
> | Method | Veh. L1 AP/APH | Veh. L2 AP/APH | Ped. L1 AP/APH | Ped. L2 AP/APH |Cyc. L1 AP/APH | Cyc. L2 AP/APH |
> | --- | --- | --- | --- | --- |--- | --- |
> | FSD (CCL, 6 epochs) | 77.3/76.9 | 69.8/69.3 | 81.4/76.0 | 73.8/68.7 | 75.8/74.5 | 73.8/72.5 |
> | FSD (SSG, 6 epochs) | 79.0/78.5 | 69.9/69.5 | 75.2/70.0 | 66.1/61.7 | 75.6/74.5 | 73.6/72.5 |
> | FSD (SSG on Veh, CCL on Ped and Cyc, 6 epochs) | 79.0/78.5 | 69.9/69.5 | 81.5/76.0 | 73.8/68.8 | 75.7/74.4 | 73.6/72.4 |
>
> The result above is very interesting.
> SSG achieves a better performance in vehicle class.
> However, the performance of the pedestrian class has severe drop. We speculate that the performance drop may be caused by the improper radius of the ball query, which is sensitive in small object detection. Multi-scale grouping might alleviate this problem. We have to leave it for future work due to time limit of rebuttal.
> Then we only adopt SSG for vehicle class, then FSD achieves better overall performance (the last row).
> We will figure out the reasons behind this phenomenon and find a better way for grouping.
> Thanks again for such a constructive suggestion!

---

> > ### Comment · Reviewer_UjRj · 2022-08-08
> > **Thanks a lot to the authors for clarifications and added and extended evaluations. This paper is very clear and the extra comparisions and analysis are very good to receive.**
> >
> > Thank you for the metric explanation, and sorry for the lack of familiarity with them. A short explanation is enough and greatly appreciated to avoid going through another link for a quick understanding. So thank you for considering adding it to the revised paper.
> > The failure case added to the supplementary material is very informative and really encapsulates the grouping mechanism problem. Looking forward for future work to see how to handle such cases.
> >
> > Thank you for the information and performance comparisons. It really shows the benefits of the authors method!
> >
> > For the shorted range and KITTI comparison, the modified table is really informative and appreciated. Of course, the time limit doesn't allow more comparisons to be done, but the short range information still gives an idea of FSD performances. Thank you for the added information.
> >
> > Concerning the multi-scale grouping, these are some very interesting findings. Of course investigating the implications of modifying the grouping algorithms to perform better on pedestrians and small object, is out of the scope of the paper and would be a paper on its own.
> > But the experiment is greatly appreciated, and I am very curious to read future work on the matter.
> >
> > Thank you again to the authors for the extra experiments and details added and explained. A very interesting paper and with a good analysis.

---

> > > ### Author Response · Authors · 2022-08-08
> > > **Sincere gratitude to the reviewer for the positive comments!**
> > >
> > > We really appreciate your positive comments, which means a lot to us!\
> > > We will definitely follow all reviewers' comments to improve our work and keep up explorations in the community!

---

### Official Review · Reviewer_2M9Y · 2022-06-28

**Rating:** 6
**Confidence:** 4
**Soundness:** 3 good
**Presentation:** 3 good
**Contribution:** 2 fair

**Summary:**

This paper studies efficient long-range LiDAR-based 3D object detection. The authors propose a fully sparse 3D object detector (FSD) to eliminate the dense BEV feature map that scales poorly with the perception range. Similar to VoteNet, FSD first applies the sparse encoder to extract point/voxel-wise features, then groups the points into instances, and finally predicts the boxes based on the instance-wise features. The authors have evaluated their proposed FSD on Waymo and Argoverse 2. FSD achieves superior performance on both benchmarks, especially Argoverse 2 where the perception range is much larger.

**Questions:**

I would love to see the answer to the following questions in the rebuttal:
* What are the pros and cons of DETR and SIR in dealing with long-range perception?
* How well does FSD scale with denser point clouds (*e.g.*, from temporal fusion)?
* Where does the efficiency boost of FSD come from?
* Could FSD still achieve the same level of speedup with highly-optimized inference engines (*e.g.*, TensorRT, TVM)?
* What is the accuracy of SST on Argoverse 2?

The authors could refer to the previous section for more detailed comments.

**Limitations:**

The authors have addressed the limitations and potential negative societal impact of their work.

**Strengths And Weaknesses:**

**[Strengths]**

This paper is well-written and easy to follow. The authors clearly explain the missing center feature problem with a good illustration (Figure 1). Generally, the figures in this paper have pretty good quality and clarity, which helps readers understand the proposed algorithm easily.

The problem studied in this paper is very important as long-range perception is very critical in high-speed driving scenarios. The proposed FSD is technically sound and achieves good empirical results on large-scale benchmark datasets.

**[Weaknesses]**

The technical novelty of this paper is a bit limited as the proposed FSD is very similar to VoteNet. The authors have clarified the differences between FSD and VoteNet in Section 3.5. However, both of them are more of engineering improvements than technical contributions. That said, the authors apply this idea to solve an important real-world problem, which could still provide valuable insights to other researchers in the community.

Researchers have spent considerable efforts on transformer-based detection heads (DETR) besides anchor-based and center-based heads. Lately, there have been several follow-ups in 3D object detection, such as Object DGCNN, DETR3D, and TransFusion. DETR head provides a potential workaround for large dense BEV feature maps. The authors could discuss the pros and cons of DETR and SIR in their rebuttal.

Although the computational cost of FSD does not grow with the perception range, it does scale linearly with the point cloud density. This could be problematic for denser point clouds (either with temporal fusion or simply with more laser beams). In contrast, dense detectors suffer less from this since the BEV feature map will always have the same resolution regardless of the point cloud size. In all experiments, the authors use single-frame point clouds for evaluation. It would be necessary to conduct experiments (or at least measure the memory footprints and inference latency) with denser multi-frame point clouds.

As this paper focuses on efficiency, it would be essential to understand where the efficiency boost comes from. For example, the authors could provide a detailed latency breakdown of different components in FSD. Besides, different inference backends could lead to very different latency results. The dense BEV encoder is composed of regular 2D convolutions, which can be easily accelerated by existing inference libraries (such as TensorRT and TVM), while the proposed SIR module is harder to be optimized. Therefore, it is unclear whether FSD could still achieve such significant speedups with highly-optimized inference engines.

Finally, there are a few aspects that could potentially be improved in the experimental evaluation:
* As FSD is based on SST, it would be necessary to report the accuracy of SST on Argoverse 2 for an apples-to-apples comparison.
* The authors could consider conducting experiments on nuScenes, where there are many competitive LiDAR baselines.

---

> ### Author Response · Authors · 2022-08-02
> **Response to R#2 (Part 1)**
>
> We sincerely thank you for the very professional comments. We are inspired and hope our discussion brings more insights.
>
>
> ## Q1: SIR v.s. DETR-like Head
> Before working on FSD, we have spent much time building fully sparse detector by replacing the dense head of SST with a DETR-like sparse head. Here we share some experiences and thoughts.
>
> (1) Directly adopting existing DETR-like heads can not make the detector fully sparse because almost all of them rely on dense feature maps. For example, in Object DGCNN and TransFusion, the sparse object queries need to gather information from dense LiDAR BEV feature maps (serve as values and keys).
>
> (2) Global cross attention on the whole scene is expensive. So the mainstream transformer-based detectors (e.g., DETR3D, Object DGCNN, BEVFormer) adopt the local deformable attention instead. However, the deformable attention requires interpolation in dense feature maps. It is not a trivial thing to do efficient differentiable interpolation in the unstructured sparse voxel features.
>
> (3) Another potential way we have tried is to apply local attention between sparse queries and their sparse neighborhood features.
> However, this raises three problems.
> (i) Due to varied object sizes, it is non-trivial to set a suitable local attention range for each query.
> (ii) The positions of queries often vary in different layers like in the deformable attention.
> The implementation of such attention between sparse features and queries with changeable positions is very tricky and hard to optimize.
> (iii) In some crowded scenes (e.g., many pedestrians), it is difficult for the queries to match objects stably and exhaustively, leading to slow convergence or a high false negative rate.
> Due to these challenges, our SST-based DETR-like detector obtains much worse performance than FSD.
>
> (4) However, the schemes proposed in (2) or (3) still have potential advantages.
> (i) Attention mechanism can be viewed as a differentiable soft grouping, which is potentially better than the grouping in FSD. We will try to adopt some learnable soft grouping strategies in future work.
> (ii) Ｔheoretically, the attention mechanism has a larger capacity than the PointNet-like SIR.
> (iii) The end-to-end manner makes it NMS-free. However, in FSD, removing NMS leads to 2 AP drops in the vehicle class.
>
> We will keep up to explore more possibilities in the future.
>
> ## Q2: Detailed Runtime Evaluation
> The evaluation is conducted on 2000 random samples from Waymo Open Dataset with 3090 GPU. We do not apply any test time optimizations. The lighter version is explained in **Q3**.
> | Component | Latency(ms, 1 frame) | Latency (ms, 3 frames)| Latency (ms, lighter, 3 frames)
> | --- | --- | --- | --- |
> | voxel feature extraction (a.k.a, VFE)| 4.0 | **16.1** | 7.0
> | Sparse Voxel Encoder (backbone)| 21.0 | 25.2 | 25.0
> | Point segmentation & voting (segmentor head)| 1.8 | **10** | 3.0
> | CCL (CPU version)| 5.3 | 7.0 | 7.0
> | Linear layer in SIR| 5.1 | 7.6 | 4.8
> | Dynamic pooling in SIR| 1.9 | 3.2 | 2.0
> | Dynamic broadcasting in SIR| 0.5 | 1.1 | 0.6
> | Group Correction| 2.6 | 5.1 | 5.0
> | Linear layer in SIR2| 4.7 | 7.5 | 5.3
> | Dynamic pooling in SIR2| 1.7 | 3.3 | 1.9
> | Dynamic broadcasting in SIR2| 0.4 | 0.9 | 0.6
> | NMS and box decoding | 7.7 | 10.2 | 8.4
> | Others | 4.1 | 6.0 | 4.7
> | Total | 60.8 | 103.1 | 75.3
>
> We also want to humbly clarify that we do not emphasize the absolute inference latency of FSD. Instead, we focus on the scalibity of FSD in the long range setting.
>
> ## Q3: Multi-frame Model
> We conduct 3-frame experiments on WOD, and we list all the results in the **shared response** to all reviewers.
> And the detailed latency is shown in **Q2**.
>
> The main latency increases from the 1-frame model to the 3-frame model lie in the widely-used VFE and segmentor head, which contain point-wise operation.
> SIR module is only applied on the foreground points, so its latency increase is acceptable.
>
> For optimization, we first use a lighter VFE and segmentation head by decreasing their channels from 64 to 32. As we discussed in Sec. 4.6 of paper, SIR could also adopt voxelization for some classes with normal or large sizes. So we apply voxelization of $ 0.3m \times 0.3m \times 0.3m $ for vehicle class for further acceleration. The performance and latency of this **lighter FSD** are shown in the **shared response** and **Q2**, respectively. The performance loss of the lighter FSD is minor.

---

> > ### Author Response · Authors · 2022-08-02
> > **Response to R#2 (Part 2)**
> >
> > ## Q4: Speedup with Highly-optimized Engines
> > As can be seen from Q2, the main source of latency is the commonly-used sparse encoder (backbone).
> > We believe there will be better support for sparse operations with the development of infrastructure.
> > For example, sparse matrix multiplication is supported in the recent version of TensorRT ([link](https://developer.nvidia.com/blog/accelerating-inference-with-sparsity-using-ampere-and-tensorrt/)).
> >
> > Some highly optimized implementations of sparse convolution are already available, like TorchSparse and Spconv2.
> > With float16 quantization, the sparse encoder could have around 45% inference speed improvement (see spconv2 [benchmark](https://github.com/traveller59/spconv/blob/master/docs/BENCHMARK.md)).
> >
> > Note that the costs of dynamic pooling and broadcast are very small.
> > The optimization of other parts is well explored since they are commonly used in many algorithms.
> > We are trying to deploy FSD in real-world applications and the speedup should be considerable.
> >
> > ## Q5: SST on Argoverse 2
> > Our devices (RTX 3090) can not afford the training memory consumption of SST on AV2 (See Fig. 5 in our paper).
> > So we adopt gradient checkpointing to save memory (leading to a very slow training speed).
> > We list the results in the **shared response**.
> > The accuracy of SST_center is lightly lower than CenterPoint mainly due to the poor performance on large objects (e.g., School Bus), caused by the center feature missing.

---

> > > ### Comment · Reviewer_2M9Y · 2022-08-09
> > > **Response**
> > >
> > > Thank you for your efforts on the additional experiments and detailed response. They have resolved most of my concerns. Therefore, I will increase my rating to 6. Great work :)

---

### Official Review · Reviewer_WF16 · 2022-07-11

**Rating:** 6
**Confidence:** 4
**Soundness:** 2 fair
**Presentation:** 2 fair
**Contribution:** 3 good

**Summary:**

This paper targets the problem of high computational complexity in LiDAR-based long-range 3D object detection. It proposes a fully sparse 3D object detection paradigm, i.e., from the feature extraction to the proposal generation and bounding box prediction, all are conducted based on sparse voxels or points. Specifically, built upon the commonly used sparse voxel encoder, it devises a novel sparse instance recognition (SIR) module to perform 3D detection based on local points, which is efficient in implementation and resolves the "center feature missing" problem. Experiments demonstrate the efficacy of this proposed method, especially for large vehicles/tiny objects and long-range cases.

**Questions:**

(Minor) I notice the model converges faster than SST and wonder whether there is any potential reason/discussion about it.

**Limitations:**

The author discusses the limitations and societal impact adequately in the paper.

**Strengths And Weaknesses:**

Strengths:
- The studied problem is valuable for practical use. The motivation that dense prediction can bring high computational complexity for long-range cases is reasonable. A fully sparse 3D object detector is suitable given the sparse property of LiDAR points distribution.
- The analysis of the difference between the proposed method and voxel-based semi-dense detectors/conventional voting in VoteNet makes the contribution of this paper clear.
- Illustrations are clear and concise, making the proposed method much easier to understand.
- Experiments support the main claims in this paper. The improvement in the detection performance of small objects/huge vehicles is impressive.
- The efficiency analysis in terms of different perception ranges shows the main advantage of the proposed paradigm.

Weaknesses:
- The methodology part is hard to follow. In particular, so many new concepts are unnecessarily defined in the presentation, for example, dynamic broadcast/pooling, sparse instance recognition, group feature aggregation, integration, sparse prediction, etc. For some of them, the author would have better expressions/names to explain them for easier understanding. There is no need to package some easy operations or commonly used modules with totally new names, which can only make the implementation hard to understand.

In contrast, the illustration is clear and concise to represent the overall pipeline and main modules of the proposed framework.

- Experiments: Although the proposed method can address the center feature missing problem to some extent and improve the detection performance of huge vehicles, it still performs a little worse than state-of-the-art methods (e.g., PV-RCNN++) on the vehicle detection of Waymo. The better mAP over 3 classes is mainly contributed by better performance on small objects. Maybe need some analysis on this phenomenon.

- The analysis of the center feature missing problem is a little vague. Specifically, is there any way that we can observe the effect of addressing this problem more straightforwardly? From my perspective, the detection performance of huge vehicles can not directly support the claim strongly, maybe at least we need a more detailed analysis of the center localization accuracy or some qualitative analysis.

---

> ### Author Response · Authors · 2022-08-02
> **Response to R#1**
>
> We are grateful for your valuable comments and constructive advice, which helps us a lot to make the paper better.
> ## Q1: Writing of Methodology Part
> Thanks for reminding! Our motivation to propose some new names is to make the concept more general and easy to describe. We will adopt plain expressions in the revision for easier understanding.
> ## Q2: Comparison with PV-RCNN++
> In the initial submission, we simply adopted the CopyPaste augmentation in previous methods (CenterPoint, SST), where the number of maximum pasted instances in each frame is 15/10/10 for Vehicle/Pedestrian/Cyclist, respectively.
> Readers could refer to the standard [config](https://github.com/open-mmlab/mmdetection3d/blob/master/configs/_base_/datasets/waymoD5-3d-3class.py#L24) in MMDetection3D.
> The SIR and SIR2 modules are trained with all foreground instances without background, which indicates FSD is sensitive to the number of pasted instances.
> As expected, we find the standard CopyPaste setting (15/10/10) is too heavy for FSD and causes overfitting in short-range vehicles and cyclists with a longer schedule.
> FSD achieves much better results after we decrease the number of pasted instances in CopyPaste augmentation (5/5/3 for vehicle/pedestrian/cyclist) and apply a longer schedule (12 epochs). For a fair comparison, we also decrease the number of pasted instances in the SST baseline, while there are no significant improvements. All the results are shown in the **shared response**.
>
> For small objects, their superior performances come from the fine-grained feature extraction in SIR. Table 6 in our paper shows that coarser representation degrades the performance of small objects.
>
> ## Q3: Analysis of Center Feature Missing
> We truly appreciate this valuable comment. We address the reviewer's concern in the following aspects.
>
> (1) First we want to humbly explain that SIR is not trying to directly overcome the center feature missing, but is a workaround for the issue. Specifically, SIR makes predictions from features of the whole instance instead of the single center feature, which might be missing or weak. We wish this intuitive explanation could help readers better understand how SIR works.
>
> (2) To compare the center accuracy as the reviewer suggested, we calculate the average center translation error (ATE) of true positives for FSD and SST_center under different vehicle length breakdowns.
>
> | Method | ATE (0m-4m) | ATE (4m-8m) | ATE (8m-12m) | ATE (12m-INF) |
> | --- | --- | --- | --- | --- |
> | SST_center| 0.144m | 0.123m | 0.323m | 0.401m |
> | FSD | 0.145m | 0.113m | 0.225m | 0.300m |
>
> (3) Following your kindly suggestions, we show extensive qualitative analysis in the **Section A of our revised supplementary materials**. Let us know if you have better suggestions, and we are happy to provide more explanations.
>
> ## Q4: Fast Convergence
> The reason is that positive samples dominate the training of SIR and SIR2 due to we segment the instances at first.

---

### Author Response · Authors · 2022-08-02
**Shared response to all reviewers: Performance Update and Qualitative Analysis**

We decrease the number of pasted objects in the CopyPaste augmentation to prevent FSD from overfitting, then we adopted a longer schedule (6 epochs -> 12 epochs) and now FSD achieves new state-of-the-art performance. We present the detailed modification in the Q2 of response to R#1.\
In the following tables, `before update` means adopting the old CopyPaste setting and `after update` means adopting the new setting.

## Waymo Open Dataset
The following table shows the updated performance as well as the results in multi-frame settings. We mark the best single-frame results in **bold**.

For a fair comparison, we also decrease the number of pasted objects for SST to update its performance. However, this strategy causes performance loss for SST, especially for the Pedestrian and Cyclist classes.

| Method | Veh. L1 AP/APH | Veh. L2 AP/APH | Ped. L1 AP/APH | Ped. L2 AP/APH |Cyc. L1 AP/APH | Cyc. L2 AP/APH |
| --- | --- | --- | --- | --- |--- | --- |
| SST_center (24 epochs, before update) | 75.4/74.9 | 66.8/66.3 | 80.3/72.3 | 72.6/65.2 | 71.6/70.3 | 68.9/67.7 |
| SST_center (24 epochs, after update) | 75.0/74.6 | 66.6/66.2 | 79.1/71.3 | 71.1/64.0 | 69.0/68.2 | 66.6/65.8 |
| PVRCNN++ (30 epochs) | 78.8/78.2 | 70.3/69.7 | 76.7/67.2 | 68.5/59.7 | 69.0/67.6 | 66.5/65.2 |
| PVRCNN++(center, 30 epochs) | 79.3/78.8 | **70.6/70.2** | 81.3/76.3 | 73.2/68.0 | 73.7/72.7 | 71.2/70.2 |
| FSD (6 epochs, before update) | 77.3/76.9 | 69.8/69.3 | 83.3/77.7 | 74.4/69.3 |73.2/71.9 | 70.8/69.6 |
| FSD (12 epochs, after update) | **79.5/79.0** | 70.3/69.9 | **83.6/78.2** | **74.4/69.4** |**75.3/74.1** | **73.3/72.1** |
| FSD (3 frames) | 80.6/80.1 | 71.5/71.1 | 85.3/81.9 | 78.5/75.2 | 80.6/79.5 | 78.4/77.4 |
| FSD (3 frames, lighter) | 80.4/80.0 | 71.4/70.9 | 85.2/81.9 | 78.3/75.1 | 80.7/79.5 | 78.5/77.6 |

We present the details of lighter FSD (the last row) in the **Q3** of response to R#2.

## Argoverse 2 Dataset
Similarly, we weaken the CopyPaste augmentation for the AV2 dataset. For clarity and simplicity, here we report the mean AP of all classes and AP of several main classes.
| Method | mean AP  | Vehicle | Bus | Pedestrian | Motorcyclist | Motorcycle | Bicyclist |C-Barrel | A-Bus | School Bus |
| --- | --- | --- | --- | --- |--- | --- | --- | --- | ---| ---|
| CenterPoint (before update)         | 22.0 | 67.6 | 38.9 | 46.5 | 28.6 | 33.4 | 20.1 | 32.2 | 8.7 | 25.8
| CenterPoint (after update)         | 22.9 | 67.1 | 39.9 | 44.5 | 33.3 | 36.9 | 23.0 | 33.0 | 10.1 | 27.8
| SST_center (before update)      | 21.4 | 65.2 | 23.3 | 50.5 | 29.4 | 37.9 | 23.9 | 34.8 | 7.0 | 20.0
| SST_center (after update)      | 22.2 | 65.0 | 24.1 | 49.3 | 33.7 | 39.4 | 24.0 | 33.2 | 8.8 | 21.0
| FSD (before update) | 24.0 | 67.1 | 39.8 | 57.4 | 30.0 | 38.1 | 27.0 | 38.1 | 15.6| 30.0
| FSD (after update)     | 28.2 | 68.1 | 40.9 | 59.0 | 39.7 | 49.0 | 33.4 | 42.6 | 20.4| 30.5

C-Barrel: Construction Barrel. A-Bus: Articulated Bus \
The updated performance on rare classes (e.g., Motorcyclist, Motorcycle) is significantly improved by alleviating overfitting.

## Qualitative Analysis
We provide extensive visualization and analysis in **Section A of our revised supplementary materials**. Let us know if reviewers have better suggestions, and we are happy to add more content!

---

> ### Author Response · Authors · 2022-08-07
> **We are open to any further discussions**
>
> Dear all,\
> Thanks again for the very constructive comments and spending your valuable time reading our responses!\
> We are open to any further discussions during the author-reviewer discussion period (Aug 2-9).

---

### Meta-Review · Area_Chair_JsFC · 2022-08-25

**Recommendation:** Accept
**Confidence:** Certain

**Metareview:**

After the rebuttal and discussion all reviewers are positive, and recommend acceptance. The AC agrees with this recommendation.

**Award:**

No

---

### Decision · Program_Chairs · 2022-09-14

Accept